# Objective hyperemia and intraocular pressure changes following omidenepag isopropyl application

Kana Tokumo[1]*, Tsuyoshi Yoneda[2], Yuta Nakaniida[3], Ryota Aoki[3], Shunsuke Nakakura[3], Taro Baba[1], Naoki Okada[1], Ayaka Edo[1], Hiromitsu Onoe[1], Hideaki Okumichi[1,4], Kazuyuki Hirooka[1], Diane Sonassa[5], Hitoshi Tabuchi[3], Yoshiaki Kiuchi[6], Hirokazu Sakaguchi[1]

1 Department of Ophthalmology and Visual Science, Graduate School of Biomedical Sciences, Hiroshima University, Hiroshima, Japan, 2 Department of Orthoptics, Faculty of Rehabilitation, Kawasaki University of Medical Welfare, Okayama, Japan, 3 Department of Ophthalmology, Saneikai Tsukazaki Hospital, Hyogo, Japan, 4 Department of Ophthalmology, Yoshijima Hospital, Hiroshima, Japan, 5 Faculty of Health Sciences and Technology, Gamal Abdel Nasser University, Conakry, Republic of Guinea, 6 Hiroshima Eye Clinic, Hiroshima, Japan

* kanatokumo@hiroshima-u.ac.jp

## Abstract

Conjunctival hyperemia was the most common adverse drug reaction to the ophthalmic solution omidenepag isopropyl in a post-marketing observational study. Here, we prospectively examined changes in hyperemia from before to 3 months after the initiation of the omidenepag isopropyl, a selective E-prostanoid receptor-2 receptor agonist, using hyperemia analysis software. Subjects were glaucoma patients who started omidenepag isopropyl use at Hiroshima University Hospital and Saneikai Tsukazaki Hospital. Hyperemia was compared by calculating the percentage coverage of vessels on the apex side using hyperemia analysis software based on anterior segment images. A total of 45 patients were enrolled, with 19 eyes in the new administration treatment group and 26 eyes in the group switched from F-prostanoid receptor agonists. Following switching from F-prostanoid receptor agonists to omidenepag isopropyl, there was no significant change in the hyperemia area from 7.4 (5.7–8.8) % to 7.4 (5.3–8.3)% (P = 0.53, Wilcoxon signed-rank test). In the new administration group, the hyperemia area increased significantly from 7.0 (5.2–8.6) % to 8.2 (6.4–9.1) % (P = 0.02, Wilcoxon signed-rank test). There was no significant change in intraocular pressure (IOP) from 15.0 (12.0–17.0) mmHg to 14.0 (12.0–15.0) mmHg in the switching group (P = 0.37, Wilcoxon signed-rank test), whereas there was a significant IOP reduction from 15.0 (13.5–18.5) mmHg to 15.0 (11.0–16.5) mmHg (P = 0.03, Wilcoxon signed-rank test) in the new administration group. One eye in the new administration group that developed macular edema during the observation period resolved spontaneously upon discontinuing omidenepag isopropyl. Both hyperemia and IOP reduction with omidenepag isopropyl ophthalmic solution were comparable to those achieved with F-prostanoid receptor agonists.

**Data availability statement:** all raw data necessary to reproduce our study results are included. We have uploaded the minimal data set. Values behind all summary statistics and graphs, plus extracted image data as Supporting Information files.

**Funding:** The author(s) received no specific funding for this work.

**Competing interests:** The authors have declared that no competing interests exist.

## Introduction

Glaucoma is the leading global cause of irreversible blindness. As populations age, the incidence of glaucoma increases, and once established, visual impairment is irreversible, exacerbating medical and economic burdens [1]. Glaucoma affected 60 million people globally in 2010, and this number was estimated to have increased to 80 million by 2020 [2]. It is estimated that there will be a global prevalence of 111.8 million by 2040 [3].

The lowering of intraocular pressure (IOP) is the only evidence-supported treatment [4,5]. Ocular hypotensive drops, laser surgery, and open surgery are listed as IOP-lowering modalities [6–9]. In open-angle glaucoma, F-prostanoid receptor (FP receptor) agonists, which are prostaglandins (PGs), are analogs that are the most commonly used first-line agents due to their best IOP-lowering effect and good acceptability in terms of frequency of eye drops and side effects [9]. FP receptor agonists are also called prostaglandin F2 alpha eye drops, with bimatoprost, latanoprost, and travoprost are among the most efficacious drugs, although the within-class differences were small and may not be clinically meaningful [10].

Omidenepag isopropyl ophthalmic solution is a selective E-prostanoid receptor-2 (EP2 receptor) agonist. It was launched in Japan in 2018 [11]. EP2 receptor agonists have a novel mechanism of action different from that of FP receptor agonists [11,12]. Omidenepag isopropyl is a prodrug. Following ophthalmic administration, it is rapidly metabolized in the cornea to its active metabolite, omidenepag, which selectively binds to EP2 receptors and stimulates aqueous humor outflow through the fibrovascular outflow and uveoscleral outflow tracts, resulting in a hypotensive effect [12].

EP2 receptor agonists' IOP-lowering effect is non-inferior to that of latanoprost, an FP receptor agonist, and EP2 receptor agonists can also be used as first-line agents [13]. FP receptor agonists have known side effects such as eyelash elongation, pigmentation, and PG-associated periorbitopathy. By contrast, omidenepag isopropyl has no such side effects [14]. The most common adverse drug reaction of omidenepag isopropyl ophthalmic solution in a post-marketing observational study was hyperemia, which occurred in 3.6% of the patients [14]. Cystoid macular edema was reported to be present in 26.9% of 52 patients with pseudophakic eyes [15], whereas this form of edema was not observed in patients with phakic eyes in a post-marketing observational study [14].

The Glaucoma Adherence and Persistency Study showed that hyperemia was the most common side effect and that it was responsible for the stopping or switching of medication in 63% of patients who stopped taking an FP receptor agonist [16]. However, hyperemia as a side effect of omidenepag isopropyl and FP receptor agonists has not been examined objectively.

In the present study, we prospectively examined changes in hyperemia from before to 3 months after the initiation of omidenepag isopropyl ophthalmic solution use by an objective method using hyperemia analysis software [17,18].

## Materials and methods

This study was approved by the ethics committee of Hiroshima University Hospital (Hiroshima, Japan) and conducted in accordance with the tenets of the Declaration of

Helsinki. Patients were enrolled between April 3, 2019 and February 21, 2022. All patients provided written informed consent before participation. The study was performed at the Department of Ophthalmology of Hiroshima University Hospital and Tsukazaki Hospital (Hyogo, Japan). This two-center clinical trial specifically enrolled patients with open-angle glaucoma or ocular hypertension who required glaucoma medical treatment and were aged >20 years.

Due to the risk of causing cystoid macular edema and associated visual impairment and vision loss [15], omidenepag isopropyl cannot be used in patients who have psudophakic or aphakic eyes. Therefore, patients with post cataract surgery or who had undergone internal ocular surgery within 1 year were excluded from the study. Additionally, patients were excluded with diseases or conditions causing conjunctival hyperemia. Patients who wore contact lenses or were undergoing eye drop therapy for diseases other than glaucoma, with a history of blepharochalasis, pterygium, allergic conjunctivitis, uveitis, or other diseases that cause hyperemia were also excluded. When both eyes met the inclusion criteria, only the right eye was analyzed. Patients who were switched from the originally used FP receptor agonist to omidenepag isopropyl eye drops were categorized into the switching group. We informed all eligible patients, not only those experiencing conjunctival hyperemia, that a new drug was now available. Patients who expressed a preference for the new therapy were then switched from FP receptor agonists to omidenepag isopropyl. Patients with no prior treatment history with glaucoma eye drops who were administered omidenepag isopropyl for the first time were categorized into the new administration group.

Goldmann applanation tonometry was performed to determine the IOP. We applied the hyperemia analysis software NIDEK (Kamagori, Aichi, Japan) developed by Yoneda et al. to determine the degree of hyperemia [17,18]. The temporal area was selected within which the evaluation was made because it has the widest area among the four conjunctival fields. The measured location and area were constant in each individual. This software measures the occupancy of the vessels in the region of interest as percent coverage (%) with conjunctival blood vessels (Fig 1) and is highly reproducible [17,18]. Slit-lamp images were obtained using an SL-D7 camera (TOPCON, Tokyo, Japan). The detailed slit-lamp photographic conditions were as follows: an angle between the slit lamp and the microscope arm of 30°; the camera flashlight adjusted to level one; a slit width of 20 mm; and an objective magnification of 10×. The diffuser of the slit lamp was used. Images were transferred to the software program for automatic pixel value calculation and converted to binary data. The percentage pixel coverage was calculated by dividing the frequency of blood vessel pixels by the total pixel frequency. IOP and temporal conjunctiva images were obtained before and 3 months after the initiation of omidenepag isopropyl administration.

## Statistical analysis

All statistical analyses were performed using JMP version 16 (Cary, NC, USA). Continuous variables are expressed as median (interquartile range) unless otherwise indicated. Changes in hyperemia and IOP from before to after the initiation of omidenepag isopropyl administration were evaluated with Wilcoxon signed-rank test. $P$-values < 0.05 denoted statistically significant differences: All statistical tests were two-sided, and $P$-values < 0.05 were considered to be statistically significant. Sample size calculations based on the results of previous studies [19] determined that 19 eyes in each group were required to detect a 3.5-pixel difference in hyperemia, with a standard deviation of 2.5 and power of 80%.

## Results

Forty-five patients (45 eyes) with open-angle glaucoma or ocular hypertension were included in the study. The patients' demographics and basic characteristics are summarized in Table 1. Twenty-six eyes were enrolled in the switching group and 19 eyes in the new administration group. In the switching group, six patients had a history of glaucoma surgery. Six patients in the switching group and two patients in the new administration group had a history of glaucoma surgery, which comprised trabeculectomy and trabeculotomy.

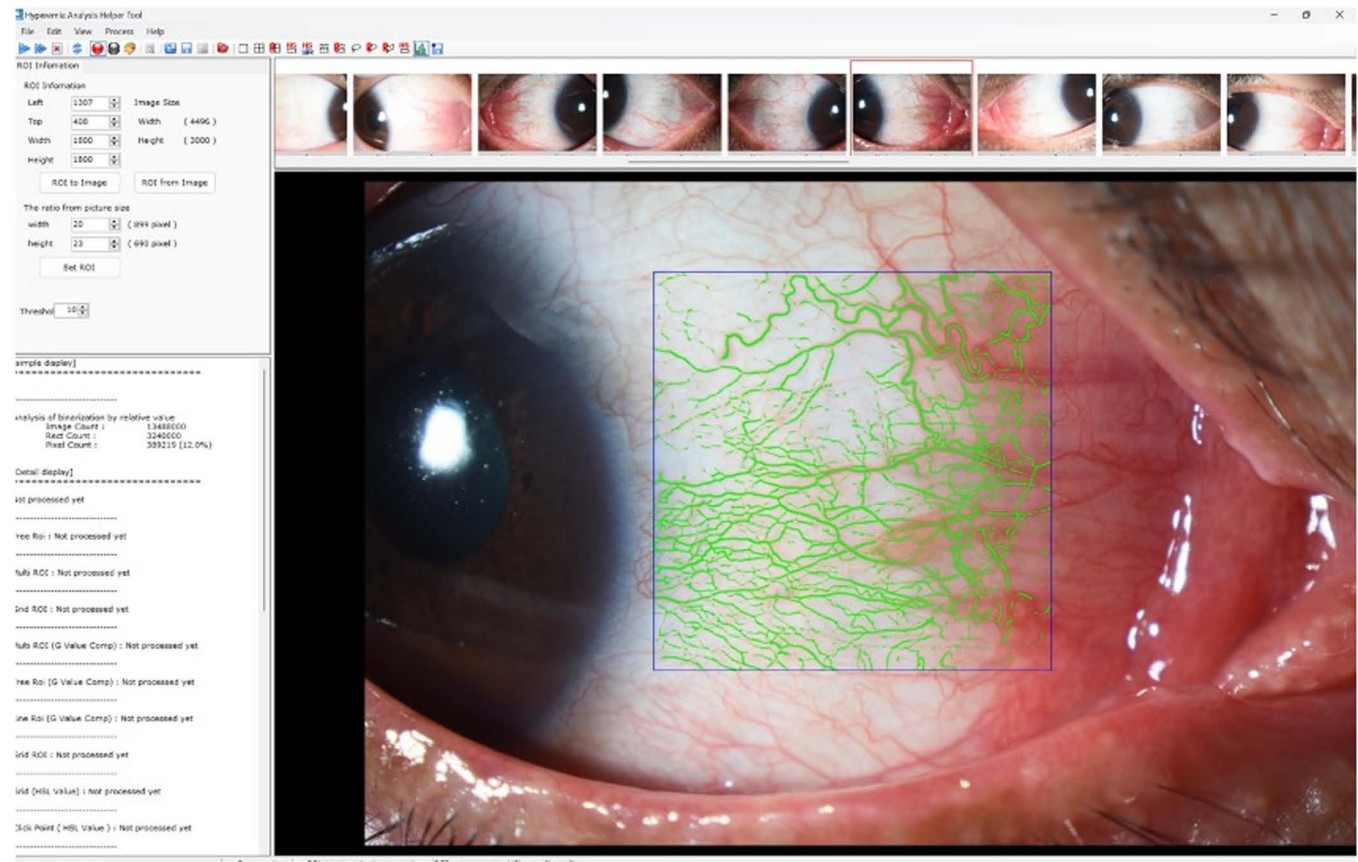

**Fig 1. Representative image using the NIDEK software program.** Using the hyperemia analysis software program, we evaluated the proportion of blood vessels automatically. The percentage of pixel coverage was calculated by dividing the frequency of the blood vessel pixels by the total pixel frequency. The blood vessel pixel count in this case is 12%.

**Table 1. Demographics and Clinical Characteristics of Patients.**

|  | Switching group | New administration group | *P*-values |
|---|---|---|---|
| Number of eyes | 26 | 19 |  |
| Age | 64.5 (57.0-71.8) | 52.0 (47.0-58.0) | 0.02* |
| Sex: Male/Female | 11/15 | 9/10 | 0.97† |
| IOP (mmHg) | 15.0 (12.0-17.0) | 15.0 (13.5-18.5) | 0.17* |
| Number of glaucoma medications | 1.5 (1.0-2.8) | 0 | <0.001* |
| Degree of hyperemia(pixel) | 7.4(5.7-8.8) | 7.0(5.2- 8.6) | 0.55* |
| Number of previous glaucoma surgeries | 6 | 2 | 0.28† |
| Types of glaucoma (eyes) |  |  | 0.35† |
| Primary open-angle glaucoma | 20 | 15 |  |
| Preperimetric glaucoma | 1 | 2 |  |
| Secondary open-angle glaucoma | 3 | 0 |  |
| Developmental glaucoma | 2 | 1 |  |
| Ocular hypertension | 0 | 1 |  |

IOP, Intraocular pressure.

\* Mann–Whitney U test, †Pearson Chi-square test.

No differences in the sex distribution, baseline IOP, or glaucoma type were found between the two groups. The age was 64.5 (57–71.8) years in the switching group, while it was significantly younger at 52.0 (47.0–58.0) years in the new administration group ($P=0.02$). The originally used glaucoma medications are shown in Table 2.

Following omidenepag isopropyl administration, there was no significant change in hyperemia from 7.4 (5.7–8.8) % to 7.4 (5.3–8.3) % ($P=0.53$, Wilcoxon signed-rank test) in the switching group (Fig 2a). By contrast, in the new administration group, hyperemia increased significantly from 7.0 (5.2–8.6) % to 8.2 (6.4–9.1) % ($P=0.02$, Wilcoxon signed-rank test) (Fig 2b).

There was no significant change in IOP from 15.0 (12.0–17.0) mmHg to 14.0 (12.0–15.0) mmHg in the switching group ($P=0.37$, Wilcoxon signed-rank test) (Fig 3a). By contrast, IOP was significantly reduced from 15.0 (13.5–18.5) mmHg to 15.0 (11.0–16.5) mmHg ($P=0.03$, Wilcoxon signed-rank test) in the new administration group (Fig 3b).

One eye in the new administration group developed cystoid macular edema during the observation period (Fig 4a). Discontinuation of eye drop use resulted in the resolution of cystoid macular edema after 1 month (Fig 4b).

## Discussion

In this study, we objectively compared conjunctival hyperemia before and after starting omidenepag isopropyl use. The new administration group exhibited significantly decreased IOP and increased hyperemia. Previous studies showed that the hyperemia of both omidenepag isopropyl and FP receptor agonist treatments was strong immediately after they were initiated, whereas after 1 month of use, the degree of hyperemia was reduced [20,21]. However, the hyperemia did not resolve; rather, it was only reduced, and chronic hyperemia remained. A post-marketing observational study of omidenepag isopropyl reported a median time to onset of conjunctival hyperemia of 12 days and a median recovery time of 25 days [14]. Arcieri et al. reported a significant increase in the hyperemia scores on latanoprost, bimatoprost, or travoprost treatment at 1 week after baseline, with peak scores observed 15 days after baseline; scores started to decrease 1 month after therapy initiation [21]. After 3 months of treatment with eye drops, stable results in reducing hyperemia were observed. In the present study, we evaluated the time point at which hyperemia and IOP stabilized at 3 months after the start of ophthalmic therapy.

Previous cross-sectional studies have suggested that glaucoma patients may suffer from a psychological burden related to the development of anxiety and depression [22,23]. Because of concerns about the possibility of blindness due to glaucoma, a younger age was thus found to be a risk factor for anxiety, while an older age and increasing glaucoma severity were risk factors for depression in patients with glaucoma [22]. It was also reported that depression is highly

**Table 2. Ocular Hypotensive Drops Used Before the Study.**

|  | Switching group | New administration group |
|---|---|---|
|  | N = 26 | N = 19 |
| None | 0 | 19 |
| PG | 14 | 0 |
| PG + β | 2 | 0 |
| PG + CAI | 2 | 0 |
| PG + α2 | 1 | 0 |
| PG+ROCK | 1 | 0 |
| PG + CAI + β | 3 | 0 |
| PG + CAI + α2 | 1 | 0 |
| PG + CAI + α2 + β | 2 | 0 |

PG, Prostanoid FP receptor agonist; β, β-blocker; α2, α2 agonist; CAI, carbonic anhydrase inhibitor; ROCK, ROCK inhibitor.

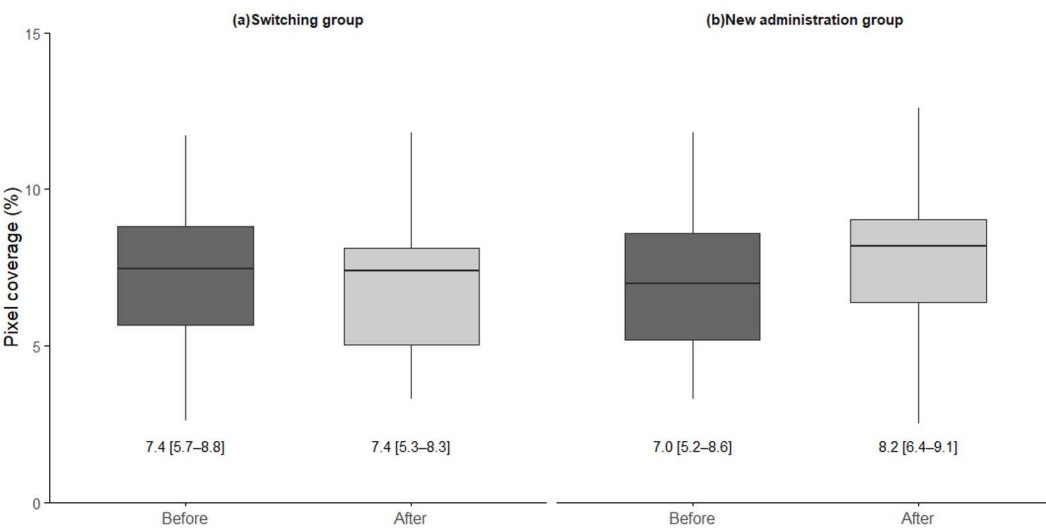

**Fig 2. Changes in hyperemia from before to after omidenepag isopropyl administration in different patient groups. (a)** Comparison of the degree of hyperemia before and after omidenepag isopropyl in the switching group. There was no significant difference in hyperemia after switching to omidenepag isopropyl eye drops ($P=0.53$, Wilcoxon signed-rank test). **(b)** Comparison of the degree of hyperemia before and after omidenepag isopropyl in the new administration group. There was a significant increase in hyperemia after starting the eye drops ($P=0.02$, Wilcoxon signed-rank test).

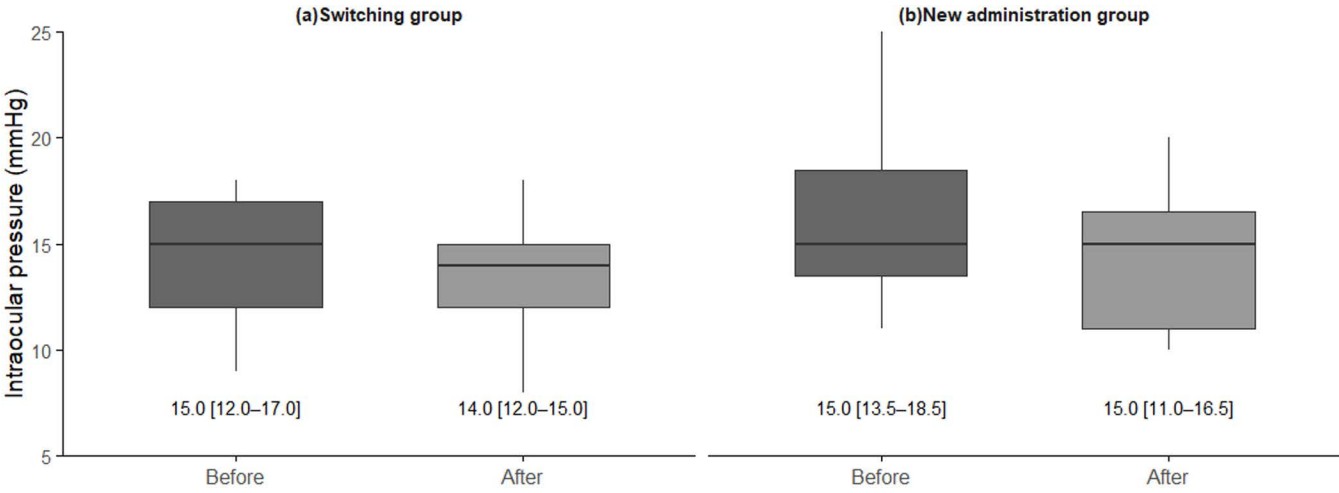

**Fig 3. Changes in intraocular pressure (IOP) from before to after omidenepag isopropyl administration in different patient groups. (a)** Comparison of IOP before and after omidenepag isopropyl in the switching group. There was no significant difference in IOP after switching to omidenepag isopropyl eye drops ($P=0.37$, Wilcoxon signed-rank test). **(b)** Comparison of IOP before and after omidenepag isopropyl in the new administration group. There was a significant decrease in IOP after starting the eye drops ($P=0.03$, Wilcoxon signed-rank test).

associated with lower rates of adherence [24]. The common side effects of topical antiglaucoma medications and the factors affecting compliance with glaucoma treatment are conjunctival injection, a stinging sensation, and blurred vision, which were also the most frequent uncomfortable side effects [25]. Conjunctival hyperemia disturbs glaucoma therapy adherence and it is a severe problem in glaucoma treatment [26]. In the present study, the side effect of hyperemia with omidenepag isopropyl was not milder than that of FP receptor agonists; rather, it was comparable.

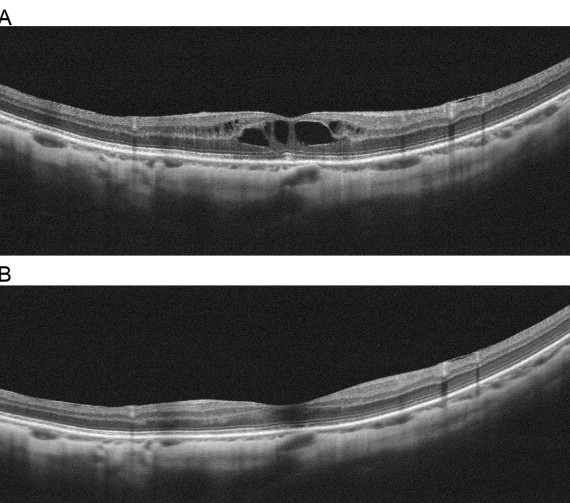

**Fig 4. Development and resolution of macular edema associated with omidenepag isopropyl administration. (a)** Macular edema appeared in the third month after omidenepag isopropyl was initiated. **(b)** Discontinuation of omidenepag isopropyl resulted in resolution of the macular edema after 1 month.

Conjunctival hyperemia results in increased medical and pharmacy costs [27]. If treatment regimens are not followed as prescribed, changes in eye drops and additional office visits may be required, leading to increased medical costs. Because of the increased burden on the patient in terms of both money and time, drugs with fewer complications such as hyperemia are desirable. Omidenepag isopropyl was comparable to FP receptor agonists in terms of hyperemia. Although the current frequency and degree of omidenepag isopropyl hyperemia is limited, symptoms vary between individuals; thus, careful observation and questioning are necessary to ensure that the medication is appropriate for the patient.

There is a high rate of cystoid macular edema when omidenepag isopropyl is instilled into aphakic eyes [15]. Therefore, omidenepag isopropyl cannot be prescribed to patients after cataract surgery. A post-marketing observational study reported a cystoid macular edema incidence of 0.2%, and all such patients had phakic eyes on both sides [14]. In our study, cystoid macular edema was observed in one phakic eye. Consequently, omidenepag isopropyl was discontinued, and the cystoid macular edema was rapidly resolved with no change in visual acuity. The exact mechanisms of cystoid macular edema on omidenepag isopropyl treatment are currently unknown [15].

It is thought that the lens capsule acts as a diffusion barrier between the anterior and posterior eye segments and maintains topographical integrity of the anterior segment [28]. Intraocular lens implantation in cataract surgery may induce changes in lens epithelial cells and other tissues of the anterior segment subjected to surgical stress and lead to endogenous cyclooxygenase enzyme 2, prostaglandin E2, and cytokine production [29,30]. Postsurgical inflammation leads to the inhibition of PG clearance from the aqueous humor by the ciliary processes, resulting in the accumulation of endogenous PG production in the anterior chamber. Accumulation of inflammatory mediators may result in blood–aqueous barrier breakdown. Transvitreal diffusion of inflammatory mediators to the retina may result in blood–retinal barrier breakdown and subsequent development of cystoid macular edema [31]. Prostaglandin E2 and prostaglandin F2 alpha have both been reported to induce inflammatory reactions in the eye [31,32]. EP2 stimulation may cause cystoid macular edema or iritis when the blood, aqueous, or retinal barrier in the treated eye has been damaged [15]. FP receptor agonists are well known to precipitate cystoid macular edema; therefore, it is essential to stop their use and maintain strict IOP control during treatment. Topical non-steroidal anti-inflammatory drugs have been shown to be an effective substitute for corticosteroids in managing cystoid macular edema—and, unlike steroids, they do not raise IOP [31].

A study limitation was the time period for the evaluation of hyperemia being limited to 3 months. Changes in hyperemia and the appearance of complications that may occur with long-term use should also be evaluated. Although the FP receptor agonists originally used in the switching group were of multiple types, including latanoprost, tafluprost, travoprost, and bimatoprost, the changes in hyperemia for the individual drugs were not evaluated. There was a significant difference in age between the switching and treatment groups. However, because this study did not compare the two groups, we believe this difference does not affect the results.

In conclusion, omidenepag isopropyl is a safe and effective glaucoma ophthalmic drug comparable to PG FP agonists. It displayed the same IOP-lowering and hyperemia effects as FP receptor agonists and had no PG-associated periorbitopathy side effects. Omidenepag isopropyl can be a first-line treatment option for glaucoma in naive patients, a switching option, and an additional option for concomitant therapy. Because it may cause cystoid macular edema, it cannot be used for aphakic or pseudophakic eyes. Furthermore, because cystoid macular edema may occur in some cases even in phakic eyes, careful follow up by optical coherence tomography is necessary.

## Supporting information

**S1 Data. All relevant data. All relevant data are available in S1 Data.**
(XLSX)

## Acknowledgments

We thank Robert Blakytny, DPhil, from Edanz (https://jp.edanz.com/ac) for editing a draft of this manuscript.

## Author contributions

**Conceptualization:** Kana Tokumo, Hitoshi Tabuchi, Yoshiaki Kiuchi.

**Data curation:** Kana Tokumo, Yuta Nakaniida, Ryota Aoki, Shunsuke Nakakura, Taro Baba, Naoki Okada, Ayaka Edo, Hiromitsu Onoe, Diane Sonassa.

**Formal analysis:** Kana Tokumo, Tsuyoshi Yoneda.

**Methodology:** Tsuyoshi Yoneda, Ryota Aoki, Shunsuke Nakakura.

**Resources:** Kana Tokumo, Tsuyoshi Yoneda, Ryota Aoki, Shunsuke Nakakura, Taro Baba, Hideaki Okumichi, Kazuyuki Hirooka.

**Software:** Tsuyoshi Yoneda.

**Supervision:** Tsuyoshi Yoneda, Yoshiaki Kiuchi, Hirokazu Sakaguchi.

**Writing – original draft:** Kana Tokumo.

**Writing – review & editing:** Kana Tokumo.

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
