## [Decision Letter · Decision Letter 0]

PONE-D-24-44578Objective hyperemia and intraocular pressure changes after application of omidenepag isopropylPLOS ONE

Dear Dr. Tokumo,

Thank you for submitting your manuscript to PLOS ONE. After careful consideration, we feel that it has merit but does not fully meet PLOS ONE’s publication criteria as it currently stands. Therefore, we invite you to submit a revised version of the manuscript that addresses the points raised during the review process.

 The manuscript is well drafted but there is a scope of improvement.

We look forward to receiving your revised manuscript.

Kind regards,

Natasha Gautam, MBBS, MS

Academic Editor

PLOS ONE

Journal Requirements:

- https://coek.info/pdf-cystoid-macula-edema-related-to-cataract-surgery-and-topical-prostaglandin-analo.html ?

In your revision ensure you cite all your sources (including your own works), and quote or rephrase any duplicated text outside the methods section. Further consideration is dependent on these concerns being addressed

3. We note that your Data Availability Statement is currently as follows: “All relevant data are within the manuscript. If additional data are needed, please contact the corresponding author, and they will be provided”

Please confirm at this time whether or not your submission contains all raw data required to replicate the results of your study. Authors must share the “minimal data set” for their submission. PLOS defines the minimal data set to consist of the data required to replicate all study findings reported in the article, as well as related metadata and methods (https://journals.plos.org/plosone/s/data-availability#loc-minimal-data-set-definition ).

If your submission does not contain these data, please either upload them as Supporting Information files or deposit them to a stable, public repository and provide us with the relevant URLs, DOIs, or accession numbers. For a list of recommended repositories, please see https://journals.plos.org/plosone/s/recommended-repositories .

Additional Editor Comments:

The authors have drafted a nice study. Were there any patients in the study for whom omidenepag isopropyl was stopped due to hyperemia, or if the patients were so symptomatic that required artificial tear drops or additional intervention?

Reviewers' comments:

Reviewer's Responses to Questions

**Comments to the Author**

1. Is the manuscript technically sound, and do the data support the conclusions?

Reviewer #1: Partly

2. Has the statistical analysis been performed appropriately and rigorously? 

Reviewer #1: No

3. Have the authors made all data underlying the findings in their manuscript fully available?

Reviewer #1: Yes

4. Is the manuscript presented in an intelligible fashion and written in standard English?

Reviewer #1: Yes

5. Review Comments to the Author

Reviewer #1: Dear authors

It is a nicely conducted study. It needs few modifications

1. Are the patients shifted from FP receptor to new drug only due to hyperemia or because of other complications of the drug?

2. Hyperemia indicates ocular surface inflammation. Table 2 shows few patients on multiple medications especially alpha agonists which can also be confounding to cause hyperemia. How do you substantiate this?

3. Are there any patients operated for glaucoma and then restarted on this new drug? Or there are any patient who have been operated for glaucoma but trab has failed and started on PGA but switched to new drug in question?

4. Table 1 shows no of glaucoma medications 1.8� 1.7 here SD is more than 1/3rd of MD which implies data is not normal , Hence need to be represented as MD with interquartile range. please redo this statistics.

5. When the age group mentioned s 61.2�16.7 how can u explain the developmental glaucoma , were they diagnosed and treated before and hence included here ?

6. Line 136, the statistic mentioned again is inappropriate as mentioned in statement 4 here. please do non parametric test and see and report.

7. Line 150 …… IOP was significantly reduced from 16.2�3.6 to 14.3�3.0. The distribution or range is overlapping how can one say it is statistically significant IOP drop?

8. Line 175 in discussion the mean onset time of hyperemia of 27�48.3 days and recovery time 32.1�37.6 days--- it is inappropriately represented.how can it be -21 days of starting medications – it is not rational.

6. PLOS authors have the option to publish the peer review history of their article (what does this mean? ). If published, this will include your full peer review and any attached files.

**Do you want your identity to be public for this peer review?** For information about this choice, including consent withdrawal, please see our Privacy Policy .

Reviewer #1: No

---

## [Author Response · Author response to Decision Letter 1]

17 May 2025

Dear Editorial Team,

Thank you for the opportunity to revise our manuscript. We have carefully addressed each of the journal’s requirements and the reviewers’ comments as detailed below.

Journal Requirements

→Thank you.We have reformatted our manuscript to fully comply with the PLOS ONE style guidelines for both the main text and file-naming conventions, using the templates provided.

https://coek.info/pdf-cystoid-macula-edema-related-to-cataract-surgery-and-topical-prostaglandin-analo.html

→Thank you for pointing out. We have rephrased all overlapping passages (Lines 228–231) and ensured that every source which is our own and others is now properly cited or quoted.

3.We note that your Data Availability Statement is currently as follows: “All relevant data are within the manuscript. If additional data are needed, please contact the corresponding author, and they will be provided.” Please confirm at this time whether or not your submission contains all raw data required to replicate the results of your study. Authors must share the “minimal data set” for their submission. PLOS defines the minimal data set to consist of the data required to replicate all study findings reported in the article, as well as related metadata and methods

→Thank you for pointing out. We confirm that all raw data necessary to reproduce our study results are included. We have uploaded the minimal data set. Values behind all summary statistics and graphs, plus extracted image data as Supporting Information files.

Additional Editor Comment

1.Were there any patients in the study who discontinued omidenepag isopropyl due to hyperemia, or who required artificial tears or additional intervention because of symptoms?

No patients discontinued omidenepag isopropyl because of hyperemia, and none required artificial tears or other interventions for ocular surface discomfort during the study period.

→Thank you for asking us. Rather than restricting the switch to those troubled by conjunctival hyperemia, we informed all eligible patients that a new drug was available. Those who expressed a preference for the new therapy were switched from FP-receptor agonists to omidenepag isopropyl (Lines 93–96).

2.Hyperemia indicates ocular surface inflammation. Table 2 shows few patients on multiple medications especially alpha agonists which can also be confounding to cause hyperemia. How do you substantiate this?

→Thank you for the question. In the switching group, only the FP receptor agonist was changed to omidenepag isopropyl; all other concomitant eye drops including alpha-agonists remained unchanged. Because we compared each patient’s hyperemia score before and after switching, any change is attributable to the FP-receptor agent switch. Statistical significance was tested with the Wilcoxon signed-rank test.

3. Are there any patients operated for glaucoma and then restarted on this new drug? Or there are any patient who have been operated for glaucoma but trab has failed and started on PGA but switched to new drug in question?

→Thank you for the question. Yes. Some patients resumed omidenepag isopropyl after prior glaucoma surgery when further IOP reduction was needed; others began postoperative PGA therapy and later switched to the new drug. The number of surgically treated eyes is now listed in Table 1, and we clarify that previous surgeries included trabeculotomy and trabeculectomy (Lines 129–131).

4. Table 1 shows no of glaucoma medications 1.8±1.7 here SD is more than 1/3rd of MD which implies data is not normal , Hence need to be represented as MD with interquartile range. please redo this statistics.

→Thank you for pointing out. We have replaced the mean ± SD with median (IQR): 1.5 (1.0–2.8) medications in the switching group.

5.When the age group mentioned s 61.2±16.7 how can u explain the developmental glaucoma , were they diagnosed and treated before and hence included here ?

→Thank you for the question. The three developmental-glaucoma patients were aged 24, 28, and 31 at enrollment. All had been diagnosed and treated in childhood; two had undergone trabeculotomy. They currently remain on topical therapy.

6. Line 136, the statistic mentioned again is inappropriate as mentioned in statement 4 here. please do non parametric test and see and report.

→Thank you for pointing out. We have replaced all parametric tests with the Wilcoxon signed-rank test where appropriate.

7. Line 150 …… IOP was significantly reduced from 16.2±3.6 to 14.3±3.0. The distribution or range is overlapping how can one say it is statistically significant IOP drop?

→Thank you for the question. We now report IOP as median (IQR):

New administration group: before 15.0 (13.5–18.5) mmHg; after 15.0 (11.0–16.5) mmHg (P = 0.03, Wilcoxon signed-rank test).

Although the ranges overlap, the paired analysis shows a significant reduction.

8. Line 175 in discussion the mean onset time of hyperemia of 27±48.3 days and recovery time 32.1±37.6 days--- it is inappropriately represented. How can it be -21 days of starting medications – it is not rational.

Response: Thank you for pointing out. The original source reported medians in addition to means, with a median time to onset of 12 days and a median time to recovery of 25 days. We rewrote the sentence in line 186-187.

We appreciate the reviewers’ constructive suggestions and believe these revisions have substantially strengthened our manuscript. Thank you for considering our revised submission.

Sincerely,

Kana Tokumo

---

## [Editor Report · Decision Letter 1]

Objective hyperemia and intraocular pressure changes following omidenepag isopropyl application

PONE-D-24-44578R1

Dear Dr. Tokumo,

We’re pleased to inform you that your manuscript has been judged scientifically suitable for publication and will be formally accepted for publication once it meets all outstanding technical requirements.

Kind regards,

Natasha Gautam, MBBS, MS

Academic Editor

PLOS ONE

Additional Editor Comments (optional):

The authors have appropriately responded to comments.
---

## [Editor Report · Acceptance letter]

PONE-D-24-44578R1

PLOS ONE

Dear Dr. Tokumo,

I'm pleased to inform you that your manuscript has been deemed suitable for publication in PLOS ONE. Congratulations! Your manuscript is now being handed over to our production team.

Kind regards,

on behalf of

Dr. Natasha Gautam

Academic Editor

PLOS ONE